# Impact of COVID-19 on Health-Related Quality of Life and Mental Health Among Employees in Health and Social Services—A Longitudinal Study

**DOI:** 10.3390/idr17060138

**Published:** 2025-11-04

**Authors:** Claudia Peters, Madeleine Dulon, Anja Schablon, Jan Felix Kersten, Albert Nienhaus

**Affiliations:** 1Institute for Health Services Research in Dermatology and Nursing, University Medical Center Hamburg-Eppendorf, 20251 Hamburg, Germany; a.schablon@uke.de (A.S.); j.kersten@uke.de (J.F.K.); 2Department of Occupational Medicine, Hazardous Substances and Public Health, Institution for Statutory Accident Insurance and Prevention in the Healthcare and Welfare Services, 22089 Hamburg, Germany; madeleine.dulon@bgw-online.de (M.D.); albert.nienhaus@bgw-online.de (A.N.)

**Keywords:** COVID-19, occupational exposure, healthcare workers, social workers, quality of life, mental health, post-COVID-19 syndrome

## Abstract

Background/Objectives: Healthcare and social workers were at increased risk of infection during the COVID-19 pandemic, and were therefore also at increased risk of long-term physical and mental health consequences due to infection. This study aimed to investigate the course of health-related quality of life (HRQoL) and mental health, in terms of depression and anxiety. Methods: A longitudinal study surveyed employees in health and social services diagnosed with SARS-CoV-2 in 2020 over a period of three years. Results: A total of 834 individuals participated in all four surveys. The mean age was 50.2 years (SD 5.8), with 82.3% of the participants being female. Mixed-model analyses were performed to examine the development over time. The results showed significant impairments in physical and mental HRQoL, as well as in mental health. Factors influencing physical HRQoL were gender, age, and pre-existing conditions. Pre-existing mental health conditions and self-reported health prior to infection were found to be predictors of mental HRQoL and symptoms of depression and anxiety. Those with persistent symptoms reported a significantly lower quality of life than those who had recovered. The mean physical HRQoL among participants with ongoing symptoms was 38.6, compared with 50.0 for those without symptoms, and the mean mental HRQoL was 40.4 versus 50.1 (*p* < 0.001). Conclusions: These findings suggest that health-related quality of life and mental health should continue to be monitored to prevent long-term psychological distress.

## 1. Introduction

Since its emergence, Severe Acute Respiratory Syndrome Coronavirus 2 (SARS-CoV-2) has been associated with substantial global morbidity and mortality. COVID-19, the disease caused by SARS-CoV-2, led to a global pandemic in 2020 and affected many aspects of daily life. The pandemic also posed major challenges in professional contexts. At the beginning of the pandemic, employees in personal service and health-related professions faced an elevated risk of infection compared to those in other occupational groups [1,2,3,4]. By the end of 2023, the Institution for Statutory Accident Insurance and Prevention in the Health and Welfare Services had received 413,077 reports of suspected occupational COVID-19 infections, of which approximately 67% had been recognised as occupational diseases. The majority of recognised cases involved employees working in residential elderly care, childcare, hospitals, and medical practices [5]. During the early waves of the pandemic, occupational factors such as contact with COVID-19 cases and the use of personal protective equipment were particularly relevant, whereas these played a less prominent role during the later Omicron waves [6].

The ongoing impact of the pandemic remains evident in the form of Long COVID and Post-COVID-19 syndromes (PCS). After infection, affected individuals have reported persistent post-viral symptoms lasting months or even years, frequently accompanied by mild to severe health impairments [7]. Long COVID is defined as the persistence of symptoms extending beyond four weeks following the acute phase of SARS-CoV-2 infection. PCS is diagnosed when symptoms persist for more than three months after the onset of infection and cannot be explained by an alternative diagnosis [8].

COVID-19 has also been linked to adverse effects on mental health. Ongoing studies conducted by the Robert Koch Institute have documented a significant deterioration in mental health within the German population during the pandemic, characterised by rising levels of depressive and anxiety symptoms. The proportion of possible depressive disorders increased from 9% in 2020 to 17% in 2022, and that of anxiety disorders from 7% in 2021 to 11% in 2022 (*p* < 0.001) [9]. Compared to the general population, healthcare workers (HCWs) showed higher prevalence rates of burnout, anxiety, depression, and stress disorders [10]. Employees involved in direct patient care were also reported to have experienced elevated and progressively increasing levels of psychological strain over the course of the pandemic [11]. Among individuals with post-COVID-19 syndrome, depressive and anxiety disorders were more common compared to those without PCS symptoms [12,13,14], accompanied by a significant reduction in health-related quality of life [13,14].

Health-related quality of life (HRQoL) is a multidimensional concept that reflects respondents’ subjective perceptions of health-related aspects of well-being and functioning [15] and goes beyond a description of their overall health condition.

To examine the longitudinal trajectory and determinants of HRQoL, a long-term study was conducted. To this end, employees in health and social care institutions who contracted COVID-19 in 2020 were surveyed. Another research question examined those factors that might influence mental health—specifically, symptoms of depression and anxiety—following a COVID-19 infection. In addition, the impact of post-COVID-19 on both outcomes was examined in more detail.

## 2. Materials and Methods

### 2.1. Study Description

Individuals insured by the Institution for Statutory Accident Insurance and Prevention in the Health and Welfare Services (BGW) who reported suspected occupational SARS-CoV-2 infections in 2020 were surveyed in an exploratory cohort study. The BGW insures employees working in non-governmental health and welfare services in Germany. In addition to healthcare professions, this includes social workers and other staff in health or social care settings. The baseline survey (T1) was conducted in February 2021, followed by T2 in October 2021, T3 in March 2022, and the final survey (T4) in April 2023 (Figure 1). The study included all insured individuals from two regions in Germany who had reported a SARS-CoV-2 infection by 31 December 2020. In total, 4325 insured individuals were invited to participate in the baseline survey, with 47% responding to the first questionnaire. The cohort, comprising 2053 individuals, was subsequently contacted for all follow-up surveys. Informed consent for participation was obtained from all subjects involved in the study. The study was approved by the Ethics Committee of the Hamburg Medical Association (2021-10463-BO-ff).

### 2.2. Variables and Measurements

A paper-and-pencil questionnaire was developed for data collection. The baseline survey included sociodemographic information, details on physical activity, smoking behaviour, height and weight, physician-diagnosed pre-existing conditions, as well as the occupational roles and workplace settings of participants. The COVID-19 infection-related questions covered parameters including the timing of diagnostic testing, details of inpatient and outpatient care, and the severity of acute symptoms categorised as mild, moderate, or severe. A detailed description is provided in Peters et al. [16]. At each survey, participants rated the severity of any ongoing symptoms as mild, moderate, or severe. Overall health status was evaluated using a numerical rating scale ranging from 0 (very poor) to 10 (very good), with assessments referring to both the pre-COVID-19 infection period and the specific time point of each survey. Scores of ≤6 were classified as poor, 7–8 as moderate, and ≥9 as good for analysis purposes. Depressive and anxiety symptoms were assessed using the Patient Health Questionnaire for Depression and Anxiety-4 (PHQ-4) [17]. The abbreviated form of the Patient Health Questionnaire (PHQ-D) utilised in this study encompasses two core diagnostic criteria for both depressive disorders and generalised anxiety disorder. Both scales range from 0 to 6, with a score of ≥ 3 indicating the presence of depressive symptoms or an anxiety disorder. The instrument demonstrates high reliability (α > 0.8). The Veterans RAND 12-Item Health Survey (VR-12) [18] was used to assess health-related quality of life. The questionnaire was adapted for the German population and is considered to be an equivalent instrument to the SF-12. The VR-12 includes 12 items covering eight domains: general health perceptions; physical functioning; role-physical; bodily pain; role-emotional, social functioning; vitality and mental health. These domains are combined into two summary scales—one for physical health and one for mental health. Each scale ranges from 0 to 100. Higher scores correspond to better health-related quality of life. The internal consistency of the sum scales can be rated as acceptable to good (α > 0.5).

### 2.3. Statistical Analysis

The analyses included participants who took part in all four surveys (n = 834) (Figure 1). Descriptive findings are presented using absolute and relative frequencies for categorical variables, and as means and distribution measures for continuous variables.

Mixed model analyses were conducted to examine changes over time. Individual participants were included as a random variable to account for their personal trajectories over time. The dependent variables analysed were physical and mental health-related quality of life as well as symptoms of depression and anxiety. Predictor variables were defined as those that are already known to be risk factors, and which were already present before the COVID-19 infection. They could therefore be used to predict the course of the outcome variables across the T1 to T4 timepoints. The models included gender, age, household composition, professional role, smoking status, obesity, physician-diagnosed pre-existing conditions, and self-rated health prior to COVID-19 infection. Results are reported as means with 95% confidence intervals (CI) and corresponding *p*-values. Initially, all available variables were included in the model. Those that were non-significant (*p* > 0.1) were then removed via stepwise backward elimination, with the exception of age and gender. Statistical significance was defined as a *p*-value below 0.05. Analyses were conducted using SPSS Version 29 (IBM Corp, Armonk, NY, USA).

## 3. Results

Of the 2053 insured individuals contacted, 834 (40.6%) took part in all four surveys, of whom 82.3% were women and 63.9% were aged 50 years or older (Table 1). Most participants worked in nursing (58.3%) or medical occupations (11.0%) in hospitals (44.2%) and residential care facilities (30.6%). Most lived in a household with adults (55.9%); nearly one-third lived with children, and 15.2% lived alone. Just under a quarter of respondents were classified as obese, and 65.2% reported having at least one physician-diagnosed pre-existing condition. Cardiovascular diseases and hormonal or metabolic disorders were the most common of these conditions. Mental health issues were reported by 10.1% of respondents. Three-quarters of respondents rated their health prior to the COVID-19 infection as good, while fewer than 3% rated it as poor. Nearly three-quarters of participants experienced severe acute symptoms during their COVID-19 infection. In total, 8.6% of insured individuals reported having received inpatient treatment. At the time of the final survey (T4), the acute infection dated back between 26 and 43 months (median duration: 34 months). At this point, 54.6% of respondents reported ongoing symptoms.

### 3.1. Changes in Health-Related Quality of Life (HRQoL) and Contributing Factors

Physical HRQoL was lower among women and participants aged 50 years and older (Figure 2). No clear trend was observed over the course of the study. Compared to other occupations, physicians reported better physical HRQoL. Participants with pre-existing cardiovascular, respiratory, or urogenital conditions reported a lower physical quality of life than those without such conditions. Mental HRQoL scores declined over the study period (Figure 3). Employment as a physician, the absence of pre-existing mental health conditions, and favourable self-rated health prior to infection were all statistically significantly associated with a higher mental quality of life.

In relation to post-COVID-19, those affected showed significantly lower HRQoL scores on both subscales compared to those who had recovered. At the time of the final survey, the mean physical quality of life score among participants with ongoing post-COVID-19 symptoms was 39.0 compared with 51.9 among those without PCS. The mean mental HRQoL score was 38.9 for participants with PCS, compared to 52.8 for those without PCS (Table 2).

### 3.2. Development and Contributing Factors in Mental Health for Depression and Anxiety

An increase in depressive symptoms over time was observed (Table 3). Statistically significant factors included occupation, pre-existing mental health conditions, and self-rated health status prior to the COVID-19 infection. The lower the self-rated health, the higher the mean score for symptoms of depression. Physicians have the lowest mean score for depression compared with other professions, particularly social care and nursing staff. A similar pattern was observed for the factors related to anxiety symptoms. Smoking also emerged as a negative predictor in this context.

Among those with ongoing post-COVID-19 syndrome (PCS), mean scores for depressive and anxiety symptoms were more than twice as high as those in individuals without PCS (Table 2). At the final survey (T4), 23.9% of respondents reported the co-occurrence of PCS with depressive symptoms, and 25.6% reported both PCS and anxiety symptoms.

## 4. Discussion

As part of a longitudinal study, the health-related quality of life of employees in health and social care facilities who contracted SARS-CoV-2 early in the pandemic was assessed at four time points over a three-year observation period. The study focused on physical and mental HRQoL as well as mental health in terms of depressive and anxiety symptoms. The results indicate that HRQoL and mental health were impaired in this cohort of workers with a COVID-19 infection. Key factors affecting physical quality of life included gender, age, obesity, and other pre-existing conditions. Mental HRQoL deteriorated over time. Pre-existing mental health conditions and self-rated health prior to the COVID-19 infection had a clear influence on both mental HRQoL and overall mental health. In addition, participants with ongoing PCS symptoms reported a considerably lower quality of life than those who had recovered.

### 4.1. Health-Related Quality of Life

Numerous studies have examined HRQoL in the context of COVID-19, but the study objectives, tools used to measure HRQoL, study populations, and follow-up periods—and thus the prevailing SARS-CoV-2 variants—varied widely, making comparisons with our results difficult. One meta-analysis reported a reduced quality of life among individuals with a SARS-CoV-2 infection compared to check-ups with follow-up periods ranging from 6 to 24 months. The infection was also associated with a high rate of impairment in activities of daily living [19]. A study in the UK examined quality of life in cases of mild to moderate COVID-19 at various time points: retrospectively before infection, during the acute phase, and during the post-COVID-19 period. The findings showed that HRQoL returned to pre-infection levels. This also applied to respondents who had been hospitalised during the acute phase of COVID-19 or had experienced post-COVID-19 syndrome [20]. In a cross-sectional study, individuals with long COVID reported the lowest quality of life and the highest level of impairment in terms of social participation compared to those who had recovered from COVID-19 or had never been infected [21].

In our study, we observed a reduced HRQoL in patients with long COVID/post-COVID for both physical (mean 38.6 vs. 50.0) and mental (mean 40.4 vs. 50.1) quality of life. Similar results have been reported in other studies. A longitudinal study of non-hospitalised individuals found that those with PCS had a lower physical (mean 46.5 vs. 52.5) and mental (mean 48.5 vs. 56.8) quality of life two years after a COVID-19 infection compared to those who had recovered [13]. The study by Naik et al. [22] also reported low HRQoL and functional impairments among people affected by PCS. By contrast, individuals of working age who had never experienced post-COVID-19 symptoms or had recovered from them had a high HRQoL. A systematic review found that PCS symptoms and ME/CFS had a considerable impact on physical quality of life and the ability to carry out daily and job-related activities [23]. Vaira et al. [24] observed reduced HRQoL caused by a persistent loss of taste and smell following COVID-19. Another study reported that individuals who later developed PCS already had lower HRQoL before their COVID-19 infection compared to a control group. In addition, a greater decline in HRQoL one year after infection was recorded among those with PCS [25]. Long COVID has also been reported to negatively affect HRQoL and work performance among healthcare workers [26]. However, the long-term study of Nehme et al. [27] found that HCWs reported lower quality of life and greater functional impairments, even early in the pandemic and regardless of a positive test result.

### 4.2. Predictors of HRQoL

In our study, age, gender, obesity and pre-existing diseases were found to have a significant impact on physical quality of life, while pre-existing mental illness and health prior to COVID-19 affected mental HRQoL. Predictors of HRQoL have also been reported by other authors with similar findings. The study by Lapin et al. [28] found that a better quality of life following a COVID-19 infection was associated with lower BMI and fewer COVID-19 symptoms. In contrast, physical HRQoL was worse in individuals with pre-existing conditions. A survey of physicians showed that pre-existing mental health conditions, higher levels of anxiety and depression, and limited professional experience were associated with a lower quality of life [29]. Among hospital staff, a lower quality of life following a COVID-19 infection was associated with obesity, not having a partner, and a lower educational level [30].

### 4.3. Mental Health

Casjens et al. [31] reported on mental distress and its increase during the pandemic among non-medical workers. Risk factors identified included a pre-existing diagnosis of anxiety or depression, chronic work-related stress, and perceived inadequate protective measures against SARS-CoV-2 in the workplace. A follow-up study conducted at four time points from 2020 to 2022 also reported a more pronounced increase in symptoms of depression and anxiety among nursing staff compared with the general population [32]. HCWs with PCS had worse scores for depression (mean 2.5 vs. 1.6) and anxiety (mean 2.4 vs. 1.4) compared to those without PCS [33]. A longitudinal study found an association between PCS and mental health impairments in non-hospitalised individuals following a COVID infection. Predictors of depressive symptoms included pre-existing anxiety or depressive disorders as well as the number of symptoms during the acute infection phase [13]. In our study, pre-existing psychological impairments and the self-assessed health status prior to COVID-19 infection in particular showed a clear effect on mental health as well as HRQoL. This should be taken into account when interpreting the study results.

### 4.4. Strengths and Limitations

In this study, insured individuals with an occupationally acquired SARS-CoV-2 infection in 2020 were longitudinally followed over a three-year period. Across institutions, a wide range of professional groups from the health and social care sectors were surveyed about the consequences of their illness at four different time points. Despite a good response rate at each follow-up, we must consider the possibility of selection bias. It is likely that individuals with ongoing symptoms and a high level of distress were more willing to participate in the study than those without post-viral issues. This may, in turn, have led to an overestimation of the prevalence of post-COVID in this cohort and should be taken into account when interpreting the results. Additionally, non-responder and loss of follow-up bias cannot be ruled out and may have influenced the results. Due to the recruitment of the participants through an accident insurance provider with a predominantly female insured population, men are underrepresented in our study. As a result, no general conclusions can be drawn with regard to gender-specific outcomes. Another limitation lies in the data collection method, which was based on a written survey. Self-reporting and self-assessment of health-impairing symptoms and conditions are subjective and may be perceived quite differently by different individuals. Objective clinical assessments were not included due to the design of the study. The absence of an appropriate comparison group without SARS-CoV-2 infection or from the general population must also be noted as a limitation, allowing no causal inference regarding the effect of SARS-CoV-2 infection. Furthermore, the influence of different phases of the pandemic, including lockdowns and various restrictions and regulations, should be taken into account when interpreting the results of our study. However, these factors were not within the scope of the study and therefore cannot be analysed in isolation.

## 5. Conclusions

Pre-existing conditions were observed as key predictors of reduced HRQoL during follow-up among employees in the health and social services following a SARS-CoV-2 infection in 2020. A pre-existing mental impairment and a self-assessed poor state of health prior to COVID-19 were found to have a particular impact on mental quality of life as well as on symptoms of depression and anxiety. More than half of the respondents indicated symptoms typical of post-COVID-19 more than three years after a SARS-CoV-2 infection. Those affected showed a considerably worse HRQoL and higher levels of depressive and anxiety symptoms. Changes in this situation and its impact on people’s ability to work should continue to be monitored to prevent long-term psychological distress. The findings demonstrate that infection risk and mental distress during a pandemic can significantly impact the health of employees in the health and social services. These suggest that, in the event of future pandemics or similar events, not only protective measures but also targeted prevention of mental strains must be considered.

## Figures and Tables

**Figure 1 idr-17-00138-f001:**
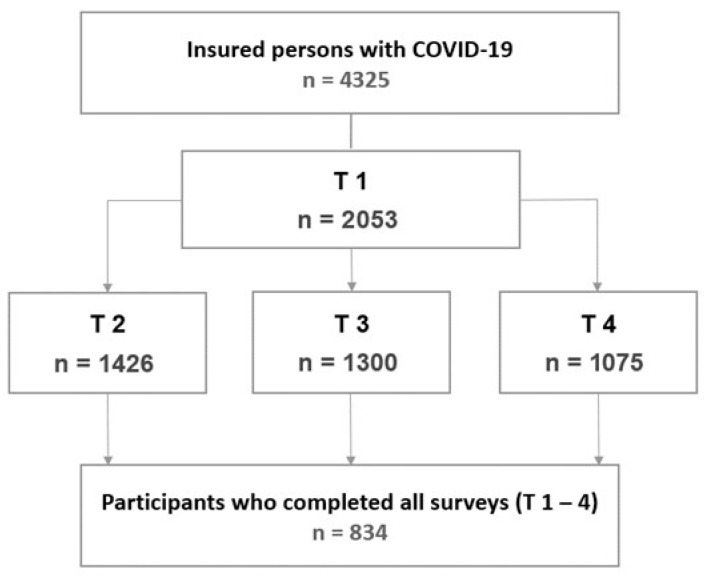
Flowchart of participants included in analysis.

**Figure 2 idr-17-00138-f002:**
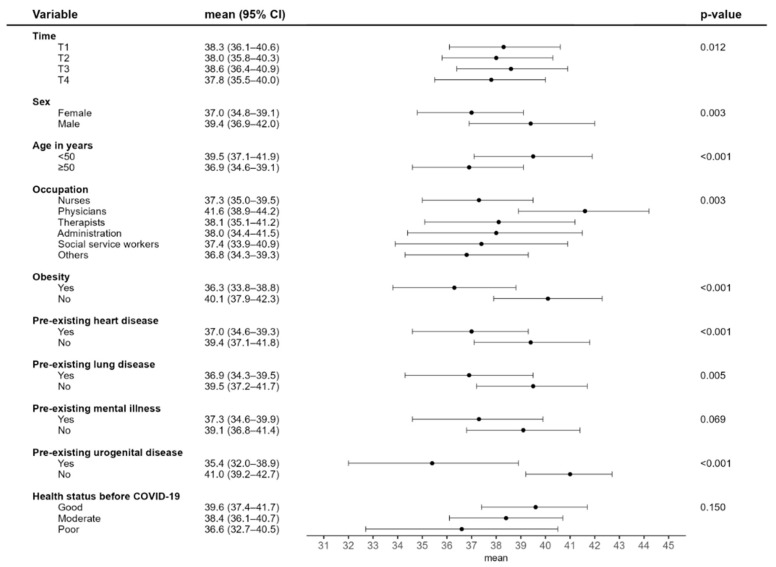
Results of the mixed model analysis for physical HRQoL.

**Figure 3 idr-17-00138-f003:**
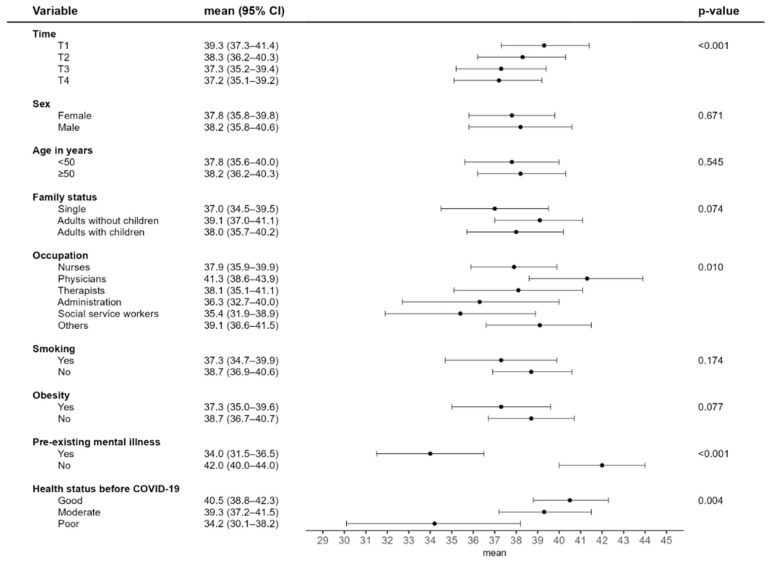
Results of the mixed model analysis for mental HRQoL.

**Table 1 idr-17-00138-t001:** Characteristics of the study population T1–T4 (n = 834).

Variable		Number (n)	Percent (%)
Sex	Female Male	686 148	82.3 17.7
Age	<50 years ≥50 years	301 533	36.1 63.9
Occupation	Nurses Physicians Therapists Social service Administration Housekeeping Others	486 92 60 36 36 32 92	58.3 11.0 7.2 4.3 4.3 3.8 11.0
Workplace N.A. = 6	Hospital Residential geriatric care Disability care Medical practice Outpatient care Other	366 253 46 43 39 81	44.2 30.6 5.6 5.2 4.7 9.8
Housing situation/ family status N.A. = 6	Adults without children Adults with children Single	463 239 126	55.9 28.9 15.2
Smoking N.A. = 5	Smoker	88	10.6
Obesity N.A. = 10	BMI ≥ 30	191	23.2
Clinically diagnosed pre-conditions		544	65.2
Pre-existing diseases	Cardiovascular disease Hormonal/metabolic disease Skin disease Respiratory disease Mental disorder Urogenital disease	243 198 112 107 84 32	29.1 23.7 13.4 12.8 10.1 3.8
Health status before COVID-19 N.A. = 9	Good moderate poor	681 122 22	82.5 14.8 2.7
Severe acute COVID-19 symptoms		612	73.4
COVID-19 hospitalisation		72	8.6
Post-COVID-19 symptoms in T4	Symptoms present No symptoms No clear classification	455 270 109	54.6 32.4 13.1

N.A.—not available/no answer.

**Table 2 idr-17-00138-t002:** Health-related quality of life and mental health in relation to post-COVID-19 at time point T4.

Time T4	All (N = 725 *)	Post-COVID-19 Symptoms (n = 455)	No Symptoms (n = 270)	
	Median, mean ± SD	Median, mean ± SD	Median, mean ± SD	*p*-value
Health-related quality of life (HRQoL)				
Physical HRQoL	43.7/42.8 ± 10.4	39.0/38.6 ± 9.6	51.9/50.0 ± 7.1	<0.001
Mental HRQoL	44.7/44.0 ± 11.7	38.9/40.4 ± 11.4	52.8/50.1 ± 9.6	<0.001
Mental Health				
Depressive symptoms	1.0/1.5 ± 1.4	2.0/2.0 ± 1.4	1.0/0.9 ± 1.1	<0.001
Anxiety symptoms	1.0/1.4 ± 1.5	2.0/1.8 ± 1.5	0.0/0.7 ± 1.1	<0.001

* Cases lacking a definitive classification of PCS were excluded from the analysis.

**Table 3 idr-17-00138-t003:** Results of the mixed model analyses for mental health for depressive and anxiety symptoms.

		Symptoms of Depression (0–6)			Symptoms of Anxiety (0–6)		
		Mean	95% CI	*p*-value	Mean	95% CI	*p*-value
Time	T1 T2 T3 T4	2.09 2.17 2.20 2.26	1.86–2.33 1.94–2.41 1.96–2.43 2.03–2.50	0.007	2.44 2.47 2.55 2.58	2.16–2.73 2.18–2.75 2.27–2.83 2.30–2.86	0.034
Sex	Female Male	2.16 2.21	1.93–2.38 1.93–2.49	0.645	2.55 2.47	2.28–2.82 2.15–2.80	0.509
Age	<50 years ≥50 years	2.15 2.21	1.89–2.41 1.98–2.44	0.502	2.44 2.58	2.14–2.74 2.30–2.86	0.127
Occupation	Nurses Physicians Therapists Administration Social service Others	2.27 1.71 2.13 2.14 2.63 2.21	2.04–2.49 1.41–2.02 1.77–2.49 1.70–2.58 2.19–3.06 1.93–2.50	<0.001	2.42 2.22 2.43 2.66 2.85 2.49	2.15–2.69 1.86–2.57 2.03–2.83 2.16–3.15 2.38–3.32 2.16–2.82	0.166
Obesity	yes no	2.27 2.09	2.00–2.54 1.86–2.32	0.066	2.61 2.41	2.29–2.92 2.14–2.69	0.063
Smoking	Smoker Non-smoker	*			2.68 2.34	2.33–3.03 2.08–2.60	0.017
Pre-existing mental illness	yes no	2.65 1.72	2.35–2.95 1.49–1.94	<0.001	3.09 1.93	2.75–3.43 1.65–2.21	<0.001
Pre-existing skin disease	yes no	*			2.60 2.42	2.27–2.93 2.15–2.69	0.152
Health status before COVID-19	good moderate poor	1.84 2.04 2.66	1.65–2.03 1.79–2.30 2.16–3.16	0.002	2.05 2.24 3.24	1.81–2.29 1.94–2.54 2.70–3.79	<0.001

* Not included in the final model.

## Data Availability

The data are available from the corresponding author upon request.

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
