# Peer review of "Impact of COVID-19 on Health-Related Quality of Life and Mental Health Among Employees in Health and Social Services—A Longitudinal Study"

_2036-7449, 2025, doi:10.3390/idr17060138_

Round 1

Reviewer 1 Report

Comments and Suggestions for Authors

First of all, I would like to congratulate the authors on conducting such a relevant and timely study. The research rigorously addresses the long-term impact of COVID-19 on health-related quality of life and mental health among healthcare and social service workers, a group particularly vulnerable during the pandemic. The choice of a longitudinal design with a three-year follow-up allows for the observation of the evolution of infection effects over time, providing significant value to the analysis of the results and strengthening the evidence on the prolonged consequences of the disease.

The use of mixed models for the analysis of longitudinal data is also noteworthy, as it adequately captures both individual variability and temporal trends. Furthermore, the identification of predictive factors such as age, gender, pre-existing conditions, and self-reported health provides valuable information for understanding the determinants of quality of life and symptoms of depression and anxiety, with clear implications for clinical practice and the continuous monitoring of physical and mental health.

As a suggestion for improvement, it is recommended to reorganize the discussion section. Specifically, the subsection currently titled Strengths and limitations should be positioned as 4.2. Strengths and limitations to maintain the structural coherence of the manuscript and facilitate readers’ navigation and understanding. This modification will allow the study’s strengths and limitations to be highlighted more clearly and systematically, enhancing the overall presentation of the work.

In conclusion, this study represents a significant contribution to understanding the long-term consequences of COVID-19 in healthcare and social service workers. Its findings not only highlight the need for ongoing monitoring of physical and mental health but also provide key information for designing preventive interventions and support policies targeted at this vulnerable group. The methodological rigor and depth of the analysis ensure that the results are highly valuable for both future research and clinical practice, reinforcing the importance of comprehensively addressing the prolonged effects of the pandemic.

Author Response

Please find the reply in the document.

Reviewer 2 Report

Comments and Suggestions for Authors

Dear Authors

I reviewed your manuscript “

 Impact of COVID-19 on health-related quality of life and mental health among employees in health and social services – A longitudinal study “ . The researcher investigated the course of health-related quality of life and mental health, in terms of depression and anxiety.  The measurement was conducted four times from February 2021 to April 2023 (3 years).    The results showed significant impairments in physical and mental HRQoL, as well as in mental health.  An increase in depressive symptoms over time was observed.  Among those with ongoing post-COVID-19 syndrome, mean scores for depressive and anxiety symptoms were more than twice as high as those in individuals without post-COVID-19 syndrome. In the final survey (2023),  the mean physical quality of life score among participants with ongoing post-COVID-19 symptoms was 39.0 compared with 51.9 among those without post-COVID-19 syndrome. The mean mental HRQoL score was 38.9 for participants with PCS, compared to 52.8 for those without post-COVID-19 syndrome.  However, it was found, also, that pre-existing mental health conditions and self-rated health before the COVID-19 infection had a clear influence on both mental HRQoL and overall mental health.

This manuscript is well-written, the data analysis was appropriate, and the data presentation was clear.   

However, I have these comments:

-In Line 48, delete this (e.g. : delete)

- The survey started in February 2021 (T1) with 2053 participants, but in 8 months, only 47% (T2, 1426) responded to participate in this survey. What were the reasons for their loss of follow-up? How can you justify this non-response?  Does it have any effect on the study results?

-In Table 3, why did the authors not include smoking in the depression model? Smoking was a significant factor for anxiety.

The study results must be interpreted with caution since pre-existing mental illness was a significant factor in the model for both anxiety and depression, and of course, QoL

-In the limitation, non-response bias and loss of follow-up bias are also important

Author Response

Please find the reply in the document.

Reviewer 3 Report

Comments and Suggestions for Authors

Thank you for the opportunity to review the study entitled “Impact of COVID-19 on health-related quality of life and mental health among employees in health and social services – A longitudinal study” (idr-3855640).

The paper presents a longitudinal study conducted in the context of the COVID-19 pandemic. The aim of the research was to investigate the course of health-related quality of life (HRQoL) and mental health, specifically in terms of depression and anxiety. A total of 834 respondents completed all survey waves (T1–T4).

In my opinion, the research topic is highly relevant, and the study is of interest. The longitudinal design and the relatively large sample size represent important strengths of the paper. In parallel, some issues need to be addressed before the manuscript can be considered for publication.

  • Abstract: Please add information about the sample (mean age, SD, percentage of men and women) to provide a clearer overview of the participants.
  • Introduction: It would be beneficial to integrate findings from trend or longitudinal studies, if available. Since the study is framed within the psychological impact of COVID-19 on individuals, I recommend including additional literature to provide a more comprehensive background. For example:
    • Hyland et al., 2021; doi: https://doi.org/10.1016/j.psychres.2021.113905
    • Gori & Topino, 2021; doi: https://doi.org/10.3390/ijerph18115651
  • Introduction: The final part of this section should be enriched with a clearer discussion of the research gap, explicitly highlighting the specific contribution of the present study.
  • Methods: The procedures for recruitment and survey administration should be described in greater detail.
  • Methods: Internal consistency indices (e.g., Cronbach’s alpha) for the current sample at each time point should be provided for all instruments used.
  • Discussion/Limitations: While the limitations are well listed, it would be important to elaborate further on how they may have influenced the interpretation of the findings (e.g., potential overestimation of post-COVID impact due to a higher participation rate of symptomatic individuals).
  • Limitations: The section should also be linked to suggestions for future research.
  • Conclusions: This section would benefit from a deeper discussion of the practical implications of the findings for healthcare and social service workers.

Best wishes,

Author Response

Please find the reply in the document.

Reviewer 4 Report

Comments and Suggestions for Authors

I would like to thank the authors and editorial staff for the chance of reading this manuscript. Overall, it adds to a growing number of studies reporting the impacts of Covid infection across the globe, albeit one of the measures used seems totally unappropriated to the population. Recommendations:

  • Please, add point estimates in the abstract. This is the second section that someone reads. As such, when you present your results, give the estimates, please.
  • Page 2, lines 54-55: please, expand and give explicit estimates for these findings.
  • Begin a new paragraph when you address a new topic (i.e., HRQoL).
  • Your definition of HRQoL is a bit vague and deserves more detail.
  • Present your goals in a separate paragraph, please.
  • Question: a measure of QoL intended for veterans was considered suitable for the study? Please include reasonable justification.
  • Present measures separately, with, at least, Cronbach’s’ alpha. CFA would be perfect.
  • The rationale for model building in the data analysis could be clearer. As it stands, it does not allow replicability.
  • Results are attractive to readers and easy to follow.
  • The discussion needs to be carefully proofread (numerous repetitions). Most importantly, when comparing it to other studies, please, include the estimates from your findings and the research being compared.
  • Finally, as I stated previously, this needs to be addressed as the choice might have been unnapropriated “Numerous studies have examined HRQoL in the context of COVID-19, but the study objectives, tools used to measure HRQoL, study populations, and follow-up periods – and  thus the prevailing SARS-CoV-2 variants – varied widely, making comparisons with our results difficult”. The only alternative I can think of for justifying the findings is to conduct CFA and SEM analyses.

Author Response

Please find the reply in the document.

Round 2

Reviewer 4 Report

Comments and Suggestions for Authors

Reliability estimates must be explicit, not just mentioning that they were high. Also, the justification for the measures intended for veterans - included in the letter - must appear in the text, so others know the basis and can build upon your study.

Author Response

Reliability estimates must be explicit, not just mentioning that they were high. Also, the justification for the measures intended for veterans - included in the letter - must appear in the text, so others know the basis and can build upon your study.

Thank you for your advice. We have included the reliability estimates and explanations for the VR-12 in the methods section.
